# A Novel Activated Biochar-Based Immunosensor for Rapid Detection of *E. coli* O157:H7

**DOI:** 10.3390/bios12100908

**Published:** 2022-10-21

**Authors:** Abdus Sobhan, Fei Jia, Lisa Cooney Kelso, Sonatan Kumar Biswas, Kasiviswanathan Muthukumarappan, Changyong Cao, Lin Wei, Yanbin Li

**Affiliations:** 1Department of Biological and Agricultural Engineering, Center of Excellence for Poultry Science, University of Arkansas, Fayetteville, AR 72701, USA or; 2Department of Agricultural and Biosystems Engineering, South Dakota State University, Brookings, SD 57007, USA; 3Department of Mechanical & Aerospace Engineering, Case Western Reserve University, Cleveland, OH 44106, USA; 4Advanced Platform Technology (APT) Center, Louis Stokes Cleveland VA Medical Center, Cleveland, OH 44106, USA

**Keywords:** electrochemical immunosensor, rapid detection, activated biochar, biosensor, *Escherichia coli* O157:H7

## Abstract

*E. coli* O157:H7, one of the major foodborne pathogens, can cause a significant threat to the safety of foods. The aim of this research is to develop an activated biochar-based immunosensor that can rapidly detect *E. coli* O157:H7 cells without incubation in pure culture. Biochar was developed from corn stalks using proprietary reactors and then activated using steam-activation treatment. The developed activated biochar presented an enhanced surface area of 830.78 m^2^/g. To develop the biosensor, the gold electrode of the sensor was first coated with activated biochar and then functionalized with streptavidin as a linker and further immobilized with biotin-labeled anti-*E. coli* polyclonal antibodies (pAbs). The optimum concentration of activated biochar for sensor development was determined to be 20 mg/mL. Binding of anti-*E. coli* pAbs with *E. coli* O157:H7 resulted in a significant increase in impedance amplitude from 3.5 to 8.5 kΩ when compared to an only activated biochar-coated electrode. The developed immunosensor was able to detect *E. coli* O157:H7 cells with a limit of detection of 4 log CFU/mL without incubation. Successful binding of *E. coli* O157:H7 onto an activated biochar-based immunosensor was observed on the microelectrode surface in scanning electron microscopy (SEM) images.

## 1. Introduction

Foodborne illness is one of the largest global health concerns that causes millions of acute illnesses, thousands of deaths, and nearly USD 15.5 billion in economic losses yearly [1,2]. *Escherichia coli* O157:H7 is one of the major food pathogens, causing around 9.4 million foodborne illnesses annually worldwide [3]. There is an abundant priority and importance for food scientists to develop an advanced method for rapidly detecting foodborne pathogens that is user-friendly, accessible, portable, and enables rapid detection of food pathogens [4]. The conventional methods used to detect foodborne pathogens such as *E. coli* O157:H7 are plate counting, indirect immunosorbent assay (ELISA), polymerase chain reaction (PCR), HPLC, biochemical test, and others [5,6,7], but the major disadvantages of these methods are that they are time-consuming (2–4 days) and laborious [8].

To minimize these barriers and technical difficulties, biosensor development has been deemed appropriate for rapid and sensitive detection of foodborne pathogens. It has various benefits over conventional assays, such as that it shortens detection time, is easy to operate, cost-effective, and portable [9]. However, there are few practical impediments to accepting existing biosensor technologies to detect food pathogens at an industrial scale due to their limited sensitivity, selectivity, and/or prohibitive cost. For example, the electrode-sensing nanomaterials, such as graphene and carbon nanotubes, used for developing biosensors may range from USD 600 to 1200/g, and the sensing systems and operations are complicated. One of the major challenges to consider when entering the market with competitive and sustainable products is the cost of sensing materials for biosensor development [8]. Furthermore, environmental safety is another critical concern for biosensor development and application [10]. Eco-friendly sensing nanomaterials should be considered in biosensor development.

Considering all these factors, biochar is a renewable and environmentally friendly material produced from pyrolysis of biomass feedstocks (e.g., wood sawdust or corn stalk) [11]. After activation, the activated biochar becomes a cost-effective (USD 0.001/g), highly porous (800–1200 m^2^/g), and electrically conductive carbon-based material with functionalized surfaces [11,12]. These unique properties of activated biochar make it an excellent candidate for sensors and/or biosensor development to overcome the challenges that remain in existing biosensors, such as high cost and environmental concerns. In addition, some researchers have previously developed the prototype of carbon electrodes using biochar as an absorbent, catalyst, or carrier. For example, Oliveira et al. and Sant’Anna et al. successfully applied biochar as a catalyst or carrier with graphite powder and mineral oil to create a sensor and/or carbon-paste electrode to detect heavy metals in functional drinks [13,14]. In this regard, biosensors developed with activated biochar have the potential to improve food quality and safety while also reducing waste significantly.

Different kinds of biosensors have been reported or are available in the market for detection of food pathogens, including optical biosensors, integrating waveguide biosensors [15], photodiode-based detection [16], or electrochemical detection systems [17], with reasonable limits of detection of foodborne pathogens. However, the majority of these biosensor assays require use of labels for foodborne pathogen detection. Electrochemical biosensors using screen-printed electrodes have been considered in this study as one of the leading techniques for monitoring foodborne pathogens for food safety [3]. They can offer several benefits over the other biosensor electrodes and substitute the conventional electrodes in biosensor devices. For example, it is easy to fabricate with nanomaterials and has the ability to use a small sample volume for testing because of having a lower surface area [18]. A variety of electrochemical biosensors with interdigitated screen-printed electrode have been demonstrated for rapid detection of pathogens, bacterial biofilm, and antibiotics [19,20,21,22]; all these biosensors showed good detection performance with a lower limit of detection. In addition, the predominant method for developing screen-printed electrodes is to immobilize biological receptors, such as antibodies, on the electrode surface. This method does not require the labeling procedure that is usually followed by other electrochemical biosensor methods and thus decreases detection time. Therefore, an imprinted electrode developed with activated biochar materials as sensing material in conjugation with antibody immobilization, called an immunosensor, has been proposed in this study as a rapid and cost-effective method to rapidly detect *E. coli* O157:H7 in foods and could be very promising in real field application. Different electrochemical biosensors with interdigitated microelectrode and electrochemical impedance spectroscopy (EIS) have been demonstrated for rapid detection of pathogens, bacterial biofilm, antibiotics, etc., such as electro-photonic traps for bacterial resistance; all these demonstrate good performance regarding detection response. Nonetheless, research on activated biochar-based electrochemical biosensors is still in its infancy. No available research has been performed yet on an activated biochar-based immunosensor for rapid detection of *E. coli* O157:H7. This research aims to develop an activated biochar-based immunosensor for rapid and accurate detection of *E. coli* cells in pure culture. Therefore, the specific objectives of this research were: (1) to activate corn stalk biochar and characterize its physical properties, and (2) to develop and characterize an immunosensor immobilized with polyclonal anti-*E. coli* antibodies to rapidly detect *E. coli* cells in culture without incubation.

## 2. Materials and Methods

### 2.1. Bacterial Preparation

Stock bacterial cultures of *E. coli* O157:H7 strain (ATCC 43888) were acquired from American Type Culture Collection (ATCC, Manassas, VA, USA). The obtained *E. coli* O157:H7 strains were stored in glycerol (20%, *v/v*) at −20 °C. Before using *E. coli* cells, the stored bacterial cells were thawed and washed with double distilled (DD) water to eliminate all glycerol. Then, the washed *E. coli* cells were suspended in Luria Bertani (LB) agar and incubated overnight at 37 °C with continuous agitation. From the overnight growing culture of *E. coli* O157:H7, 20 mL of heat-killed *E. coli* O157:H7 was followed by the previously developed method [3]. Briefly, 20 mL of overnight *E. coli* culture was treated at 80 °C for 15 min. After heat treatment, the killed *E. coli* cells were serially diluted into phosphate buffer saline (PBS, pH 7.4, 0.1 M, Thermo Scientific™, Waltham, ME, USA). The concentrations of each heat-treated dilution were confirmed by comparing the results after plating live *E. coli* cells on Trypsin soy agar (TSA). The heat treatment used to kill the *E. coli* cells did not influence recognition of the *E. coli* cells by the anti-*E. coli* pAbs, which was confirmed by dot-blot tests. The PBS solution without microbial cells was considered as the control. 

In a similar manner, stock bacterial cultures of *Listeria monocytogenes* (ATCC 43251), *Staphylococcus aureus* (ATCC 27660), *S. typhimurium* (ATCC 14028), and others were acquired from American Type Culture Collection (ATCC, Manassas, VA, USA) and used for testing the biosensor’s specificity. A solution containing streptavidin (Molecular Probes Inc., Eugene, OR, USA) was diluted at the concentration of 20 mg/mL [3]. Biotinylated labelled anti-*E. coli* polyclonal antibodies (pAbs), with a concentration of 4.0–5.0 mg mL^−1^, were obtained from Biodesign International (Saco, ME, USA) and diluted to one-hundredth-fold with carbonate-bicarbonate buffered at 12,000 rpm for 20 s. The DD water (18.2 MΩ cm) was acquired from Millipore (Milli-Q, Bedford, MA, USA). All other reagents used were of analytical grade and used without further purification.

### 2.2. Preparation and Activation of Biochar from Corn Stalk

Biochar was prepared from field corn stalk, which was locally collected from corn fields (Brookings, SD, USA) by following proprietary reactors using the accurate control pyrolysis method [23,24]. Briefly, 20 kg of corn stover from corn stalks was transferred to the reaction chamber of proprietary reactors and pyrolyzed between 250 and 300 °C. After pyrolysis, the corn stalk biochar was formed and shifted from the reaction chamber. To activate the biochar, the corn stalk biochar produced was activated in a steam reactor by following the steam activation protocol [17]. To begin with, the corn stalk biochar was finely ground using the omni mixer homogenizer (Waterbury, CT, USA). Afterward, the fine ground biochar was washed with double distilled water by centrifugation at 10,000 rpm for 10 min. Thereafter, the biochar-containing supernatants were separated and dried in an oven between 80 and 100 °C overnight. Next, the dried biochar powder was placed into the reactor bed and continuously activated at 800 °C with a steam stream and nitrogen gas for 1 h. The bed temperature of the reactor was measured by positioning a K-type thermocouple in the center of the reactor. Nitrogen flow as the carrier gas to the reactor chamber was monitored by a mass flow controller at 200 CCM, and steam with 2 mL/min via a steam chamber was purged to the reaction chamber throughout the biochar activation process. The temperature of the reactor furnace and boiler was controlled by the temperature controllers of the reactor, and an ice bath was employed to condense unreacted steam. A schematic diagram of steam activation treatment for corn stalk biochar is shown in Figure 1.

### 2.3. Characterization of Activated Biochar 

#### 2.3.1. Raman Spectroscopy Analysis 

Raman infrared (IR) spectroscopy analysis is one of the widely known spectroscopic techniques used to identify the chemical, structural, and functional properties of sample molecules [25]. To identify and compare chemical and structural changes of the corn stalk biochar before and after their steam activation, IR spectra of the sample were recorded using IR spectroscopy (LabRAM HR, HORIBA Scientific, Edison, NJ, USA), which was operated between 1000 and 2000 cm^−1^ with a polarized light wavelength of λ = 532 nm.

#### 2.3.2. Brunauer–Emmett–Teller (BET) Analysis

Characterization in terms of BET surface area of the obtained activated biochar was performed by employing a Surface Area and Porosity Analyzer (Micromeritics ASAP 2020, Norcross, GA, USA). For this analysis, activated biochar samples were first degassed overnight at 110 °C under vacuum conditions to eliminate the remaining moisture prior to N_2_ adsorption. Afterward, the curves of N_2_ adsorption obtained for each activated biochar sample were applied to calculate the specific BET surface area of the samples.

#### 2.3.3. Apparatus and Electrode Measurement System

The configuration of the interdigitated electrode, measurement system, and their electric connections is depicted in Figure 2. The bare interdigitated electrode (Aibit Tech., Jiangyin, China) was made up of six pairs of gold finger electrode. The finger width of each gold electrode was kept constant at 200 µm and the space between the two finger electrodes was 200 µm. The total area of the gold finger electrodes was about 12.38 mm^2^, and the ratio between gold finger electrodes and whole circular electrodes was 0.54:1.

Electrochemical analysis of the electrode was performed via a linear sweep voltammetry program using the electrochemical workstation (CHI 430A, Austin, TX, USA) to quantify the slope of current/voltage values from 0 V to 0.1 V. An IM-6 Impedance Analyzer (HIOKI IM3590, Melrose, MA, USA) was used to measure impedance amplitude and phase angle in the frequency range of 10–10,000 Hz regarding a sinusoidal AC signal and an amplitude of 5 mV.

### 2.4. Optimization of ABC Concentration 

Prior to optimization of the sensor’s electrode, gold (Au)-coated microelectrodes were first ultrasonically cleaned in acetone and then followed by washing with methanol and DD water. Afterward, the microelectrodes were dried in the presence of nitrogen N_2_ and kept in a vacuum chamber right after measurements. The electrical conductivity of the microelectrodes was tested to confirm that no electrical current over the electrode surface was present before sensor fabrication. The electrical conductivity of the Au-coated electrode was also assessed to verify that Au-coated electrodes were able to work effectively when each microelectrode was tied to an electrical tester. To optimize activated biochar-based electrodes, the different concentration of activated biochar powder, which was ranged between 5–40 mg/mL, was individually suspended in the *N*,*N*-dimethylformamide (DMF) and then sonicated for at least 2 h in a water-filled sonicator at room temperature. After being sonicated, 5 µL of sonicated activated biochar was dropped onto the center of the gold-plated electrode surface by drop-casting and wrapping each space between the electrodes. Next, the coated electrodes were incubated in a drying oven at 80 °C for 15 min to eliminate DMF residues and allow activated biochar to align within the electrode gaps. The resistance values (kΩ) of each concentration of activated biochar used for biosensor development were measured using a linear sweep voltammetry program.

### 2.5. Stepwise Fabrication of the Immunosensor

Fabrication of the immunosensor included three steps, which are referred to as alignment, annealing, and functionalization [26]. Alignment of activated biochar was accomplished by depositing a drop of 5 μL of optimized activated biochar concentration on the microelectrode gap and applying AC electrophoresis for a few seconds at 4 MHz until a resistance of about 1–10 MΩ was achieved. Afterward, the aligned immunosensor was annealed for 15 min in the incubator to align activated biochar between the electrode gaps to obtain good contact. Later, the annealed sensor surface was washed with DD water to remove unassembled activated biochar from the biosensing surface. To perform non-covalent functionalization, 25 µL of streptavidin as a linker solution was dropped to the washed annealed electrodes and then dried for 2 h in the open air at room temperature. Following washing steps with DD water, the developed biosensor was covalently functionalized with the corresponding pAb (anti-*E. coli*) onto the linker-modified biosensor surface and then incubated overnight in a refrigerator at 4 °C.

To detect *E. coli* cells, an aliquot of 25 µL of serially diluted *E. coli* culture was applied to the covalently functionalized electrode for 30 min at room temperature to allow antibody–antigen reactions. After the reaction, the biosensor was washed with DD water to reduce non-specific binding effects, and the resistance values and impedance amplitude of the immunosensor were measured with the linear sweep voltammetry and impedance analyzer to confirm that the change in resistance signal was due to the reaction of *E. coli* cells with pAbs. Similarly, 25 μL of PBS buffer was used to serve as a control sensor. The schematic diagram of the activated biochar-based immunosensor is presented in Figure 3. The resistance difference (∆R) was then calculated using the following Equation (1)

∆R = (R_1_ − R_0_)/R_0_,
(1)

where R_0_ is the resistance measured with antibody and R_1_ is the resistance measured with *E. coli* cells using the biosensor.

### 2.6. Antibody Conformity Test Using Dot-Blot Analysis

The conformity test of anti-*E. coli* pAbs to *E. coli* O157:H7 was performed according to previously described methods [27]. For this test, *E. coli* O157:H7 and the other non-targeted pathogens, such as *Listeria*, *Campylobacter,* and *Salmonella* species (10^8^ CFU/mL), in heat-killed forms were used for screening. Briefly, 2 μL of each bacterial cell were individually spotted onto a nitrocellulose membrane at the center of the grid (approximately 3–4 mm diam), which was exposed to dry at room temperature for 10 min. After that, the membrane strip was soaked into 5% BSA in TBS-T (Tris-buffered saline with Tween-20) for 1 h at room temperature to block the non-specific-sites. Then, the blocked membrane strip was incubated with anti-*E. coli* antibody solution for 1 h at room temperature. Following this, the antibody-conjugated membrane strip (pAb-strip) was washed with TBS-T solution for 5 min. After washing as above, the pAbs-strip was again incubated with alkaline phosphatase secondary antibody solution, diluted to 1:1500, and incubated at room temperature for 1 h. After a washing step, the strip was observed under Odyssey^®^ XF Imaging System (LI-COR Biosciences, Lincoln, NE, USA). An observed black spot at the membrane strip where the anti-*E. coli* pAbs spotted was considered a positive result. However, the absence of any black spot on the membrane strip was considered a negative result. 

### 2.7. Scanning Electron Microscopy (SEM)

The surface morphology of the activated biochar, surface of the linker-modified sensor, and *E. coli*-bound sensor were observed using a scanning electron microscope (FEI XT NOVA, Hillsboro, OR, USA). To observe the surface, sensor specimens were first coated with Au layers using a sputter-coater prior to SEM observation. The overall surfaces of the Au-coated immunosensor were then scanned with an SEM under an electric voltage of 10 kV at a working distance of 3 mm with 1500× *g* magnification.

### 2.8. Statistical Analysis

A randomized design of experiment was conducted throughout the experiments. Analysis of data was statistically performed by one-way analysis of variance (ANOVA) via sigma plot software (Version 14.0, Sigma plot, Chicago, IL, USA). The differences between means were evaluated using Tukey test with a defined significance level of *p* < 0.05. 

## 3. Results

### 3.1. Physical Properties of Activated Biochar Sample

To characterize the physical properties of the activated biochar material for immunosensor development, the IR spectrum of activated biochar was analyzed using Raman spectroscopy. As shown in Figure 4A, two high-intensity peak bands (I_G_/I_D_) were produced corresponding to the I_G_ band at 1590 cm^−1^ and I_D_ band at 1380 cm^−1^. The I_D_ band relates to sp^3^ hybridized carbon, whereas the I_G_ band corresponds to the in-plane vibration of sp^2^ hybridized carbon [28]. The intensity ratio of I_D_/I_G_ was found to be 0.85 and 0.97 for biochar after pyrolysis and activated biochar after steam activation, respectively. The I_D_/I_G_ for activated biochar was higher than for non-activated biochar, indicating more vacancy defects were produced after biochar activation [29]. The BET surface areas of biochar and activated biochar are presented in Figure 4B. It shows that the BET surface area of activated biochar was greater compared to the BET surface of non-conductive biochar: 825.89 m^2^/g of activated biochar versus 200 m^2^/g of non-conductive biochar. The large surface area of activated biochar can contribute to more binding sites to capture the targeted cells [30]. This enhanced surface area might have occurred due to elimination of carbon mass and extra volatile carbonaceous matter remaining on the biochar surface during the activation process [11]. To compare the electrical conductivity before and after biochar activation, a biosensor was developed with both biochars and characterized their electrical properties with linear sweep voltammetry. As shown in Figure 4C, the current obtained for activated biochar was significant, but no current was found for the biosensor developed by non-activated biochar.

### 3.2. The Optimum Concentration of Activated Biochar on Immunosensor Development

To optimize the concentration of activated biochar to fabricate the immunosensor at room temperature, different reasonable concentrations ranging from 5 mg/mL to 40 mg/mL were considered and fabricated sequentially on the sensor surface by drop casting, and the resistance values of each selected concentration were determined as shown in Table 1. When the activated biochar concentrations used for biosensor development were increased from 5 to 40 mg/mL, the resistance values of the developed biosensor initially increased and then decreased to 0.98 kΩ. This result was in good agreement with that of the carbon-nanotube-based biosensor (CNT-based biosensor), where the resistance values of the biosensor decreased with increasing CNT concentrations on the biosensor [31,32]. Evidently, the resistance value increased to 4.86 kΩ in the 5 to 20 mg/mL concentration range. On the contrary, in the concentration range of 20–40 mg/mL, the resistance values decreased and were less than 1 kΩ between 30 and 40 mg/mL in concentration, which denotes unstable binding of activated biochar to the sensor surface. Furthermore, the overall resistance values also fluctuated depending on the washing steps. This increasing or decreasing fluctuation in resistance values after washing steps might be attributed to unstable binding of activated biochar on the surface of biosensor electrode. However, no detectable differences were observed in the resistance values after washing steps while the activated biochar concentrations were at 20 mg/mL. The minimum concentration range was determined to be 20 mg/mL among the selected concentrations and, more importantly, offered stable resistance values even after five washing steps. Hence, the optimum concentration of activated biochar for connecting each gold electrode for immunosensor development was determined as 20 mg/mL in this study.

### 3.3. Detection Responses of Activated Biochar-Based Biosensor

The detection responses of the activated biochar-based immunosensor were performed with and without functionalization and immobilization of streptavidin as a linker and anti-*E. coli* antibodies (pAbs), respectively. As observed in Figure 5A, without functionalizing the linker and immobilizing the antibodies to the immunosensor, the resistance difference (ΔR) for the applied bacterial cells was similar compared to the ΔR of PBS. It confirmed that only the activated biochar-based immunosensor could not bind *E. coli* cells on the sensor surface. Similarly, the ΔR of the immunosensor fabricated with activated biochar and functionalized with the linker was lower and identical. It is interesting to note that, when the functionalized immunosensor was further immobilized with anti-*E. coli* pAbs, the ΔR of the immunosensor was significantly increased when compared to PBS. Based on these phenomena, several facts can be considered, such as activated biochar-based biosensors require functionalization of the linker to bind the immobilized pAbs on the sensor platform. This is because ester groups of linkers interact with the primary and secondary amine groups of pAbs via non-covalent couplings [33]. In addition, the applied bacterial cells of *E. coli* on the immunosensor responded to the amino groups of the antibody; thereby, the ΔR of the immunosensor rose. This effective coupling of pAbs in capturing bacterial cells was thought to raise the resistance value significantly. It also described that the rise in ∆R of the biosensor was ascribed to accumulation of negative charge from the antibodies.

Figure 5B represents an electrochemical impedance amplitude obtained for detection of *E. coli* cells. As shown in Figure 5B, when the immunosensor is immobilized with anti-*E. coli* pAbs overnight and incubated with *E. coli* cells, the impedance amplitude significantly rose compared to the immunosensors that were fabricated with only activated biochar and activated biochar plus linker. It is suspected that the binding of pAbs to *E. coli* O157:H7 inhibited transfer of electrons with respect to an applied frequency of 10–10,000 Hz over the sensor, indicating that the dielectric capacitance of the immunosensor significantly affected the current transfer and thereby increased the impedance amplitude. In addition, when the activated biochar-based immunosensor was functionalized with the linker, the impedance amplitude slightly increased from 4.9 to 5.5 kΩ. This is because the hydrophobic ends of streptavidin as a linker are being irreversibly adsorbed into the hydrophobic sidewall of an activated biochar via π–π stacking interactions [34], resulting in the impedance amplitude increasing. When spectra of the phase angle for impedance measurements were compared, the magnitude of the phase angle mostly remained constant in the low frequency range of 10–2000 Hz (Figure 5C). The phase angle values showed a significant decrease when the immunosensor immobilized with pAb reacted to *E. coli* cells. There is no noticeable phase angle change observed between activated biochar and the linker-antibody-immobilized immunosensor, which stayed close to −0.4.

### 3.4. Binding Specificity of pAbs toward E. coli O157:H7

The dot blot test is a widely used method to verify the specific reactivity of antibodies to the desired antigen [35]. As shown in Figure 6A, *E. coli* O157:H7 only reacted to the anti-*E. coli* pAb by producing black dots. The other microorganisms, including *S. aureus*, *S. typhimurium*, *L. monocytogenes*, *L. innocua*, *C. jejuni*, and PBS, were not specific enough and did not produce any black dots. Depending on these antibody specificity tests, the anti-*E. coli* pAb was deemed suitable for development of an immunosensor to detect *E. coli* cells in pure culture. To verify the specificity of the developed immunosensor, the immunosensor was assessed with various food pathogens, including *S. typhimurium*, *S. aureus*, *L. innocua*, *L. monocytogenes*, and PBS. The minimum ∆R signal was noted for nontarget pathogens, such as *S. aureus*, *S. typhimurium*, *L. monocytogenes*, *L. innocua*, as well as PBS as the control. However, as shown in Figure 6B, the largest ∆R signal was obtained for *E. coli* O157:H7, which was the target food pathogen, indicating that cross-reaction between *E. coli* O157:H7 antigens and anti-*E. coli* pAb occurred. The pAb-modified immunosensor had high binding specificity for the intended target pathogens. Therefore, it was confirmed that the activated biochar-based immunosensor immobilized with anti-*E. coli* pAbs owned excellent performance to selectively detect *E. coli* O157:H7, discriminating other bacterial cells other than the target *E. coli* O157:H7.

### 3.5. Limit of Detection of the Activated Biochar-Based Immunosensor

Electrochemical immunosensors for rapid detection of food pathogens are the most widespread class as they are economical and have a fast response time. As shown in Figure 7, the ∆R values increased when the applied concentration of bacterial cells to the developed immunosensor increased between 10^4^ and 10^7^ CFU/mL (Figure 7). This similar phenomenon of resistance change was observed for the CNT-based biosensor when the binding of pAbs with bacteria cells increased the resistance [36]. The linear regression of ∆R is presented in Figure 7. The regression coefficient (R^2^) value of the activated biochar-based biosensor was calculated as 0.85. However, in order to confirm that the observed ∆R increase was the result of binding of pAbs with *E. coli* cells and not non-specific binding of the *E. coli* cells to the activated biochar, a test was performed in which the response of activated biochar functionalized with only linker instead of pAbs. PBS as the control biosensor was considered, and the sensor did not exhibit any significant ∆R changes, suggesting that there was no specific binding of *E. coli* O157 to the activated biochar-based platform. In addition, ΔR was significantly different when the concentration of *E. coli* O157:H7 was 10^4^ CFU/mL or greater (*p* < 0.05). The activated biochar-based immunosensor could not detect *E. coli* cells less than 10^4^ CFU/mL with a detection time of 30 min. Moreover, in order to reduce the limit of detection (LOD), the key task is development of a newer enrichment process to detect a minor number of bacterial cells less than 10^4^ CFU/mL in the dilution without incubation. As this was the first study on activated biochar-based immunosensor for rapid detection of *E. coli* cells in pure culture, regarding future research, more studies are required for reduction of LODs for rapid detection of *E. coli* cells in pure culture.

### 3.6. Scanning Electron Microscopy (SEM)

SEM was used to observe the surface properties of the activated biochar-based immunosensor and verify the binding efficacy of *E. coli* to immobilized anti-*E. coli* antibodies. Figure 8A shows the surface morphology of the screen-printed electrode modified with activated biochar at a 2 mm scale. The high-resolution SEM of activated biochar was presented in Figure 8B. The activated biochar functionalized with streptavidin as a linker is demonstrated at a 20 µm scale in Figure 8C. The ester groups of linkers possess high binding affinity and are added onto the surface of the activated biochar to form a compact surface, which creates effective ester–biochar interactions. Due to the hydrocarbon properties of activated biochar, negatively charged streptavidin as a linker is freely adsorbed onto the activated biochar-modified sensor surface via electrostatic and hydrophobic interactions. In Figure 8D, the surface of antibody-immobilized immunosensor with PBS (or without *E. coli* cells) was observed under SEM at a 10 µm scale. When the immunosensor was immobilized with anti-*E. coli* antibodies, *E. coli* cells effectively immune-reacted and were able to bind *E. coli* cells significantly. Therefore, attachment of *E. coli* cells onto an activated biochar-based surface was inspected (Figure 8E,F). It confirmed that the immunosensor immobilized with antibodies was enough to capture *E. coli* cells, suggesting that specific binding interactions occurred between *E. coli* cells and the immunosensor. Many studies have focused on successful immunosensor development for microorganism detection [6,31,37], even for some other immunosensor applications in food products [38,39]. Nevertheless, to the best of our knowledge, this research presents the first activated biochar-based immunosensor developed for detecting *E. coli* cells in pure culture.

## 4. Conclusions

In this research, corn-stalk-activated biochar was generated using our proprietary reactors and served an extensive surface area of 825.89 m^2^/g. The activated biochar-based immunosensor was developed for label-free, rapid, and sensitive detection of *E. coli* O157:H7. The specificity of the developed immunosensor ensured that the anti-*E. coli* pAbs were specific enough to rapidly respond to *E. coli* cells rather than non-targeted food pathogens. The sensitivity of the immunosensor was confirmed by monitoring the changes in ∆R and impedance amplitude after bacterial cells bound to the sensor surface. A linear increase in ∆R was observed when *E. coli* concentrations grew logarithmically from 10^4^ to 10^7^ CFU/mL. A limit of detection of 10^4^ CFU/mL without incubation was achieved for detection of *E. coli* O157:H7. Microstructures of the developed immunosensor were observed with SEM, confirming that *E. coli* O157:H7 cells were captured on the electrode immobilized with antibody. We assume that the developed immunosensor will offer a promising approach for rapid detection of foodborne pathogens in foods and, furthermore, can be integrated into a portable multiplexed device for use in food industries.

## Figures and Tables

**Figure 1 biosensors-12-00908-f001:**
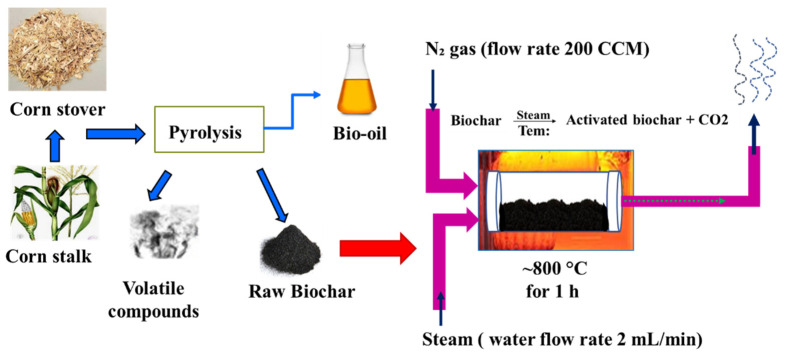
Formulation of corn stalk biochar through proprietary reactors and activation followed by steam activation treatment.

**Figure 2 biosensors-12-00908-f002:**
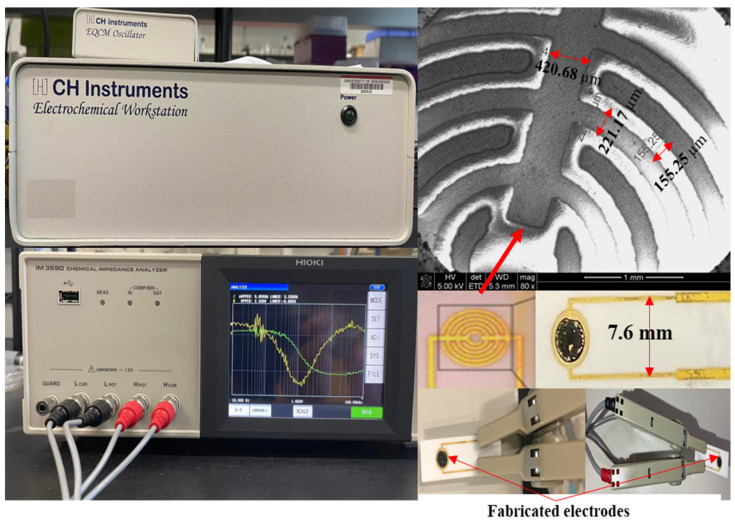
Scheme of whole systems for electrochemical workstation and impedance analyzer, including the fabricated electrode.

**Figure 3 biosensors-12-00908-f003:**
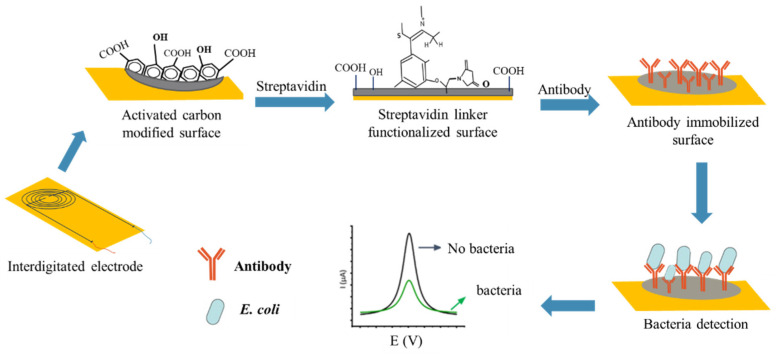
Stepwise developmental process of the activated biochar-based immunosensor to rapidly detect *E. coli* cells.

**Figure 4 biosensors-12-00908-f004:**
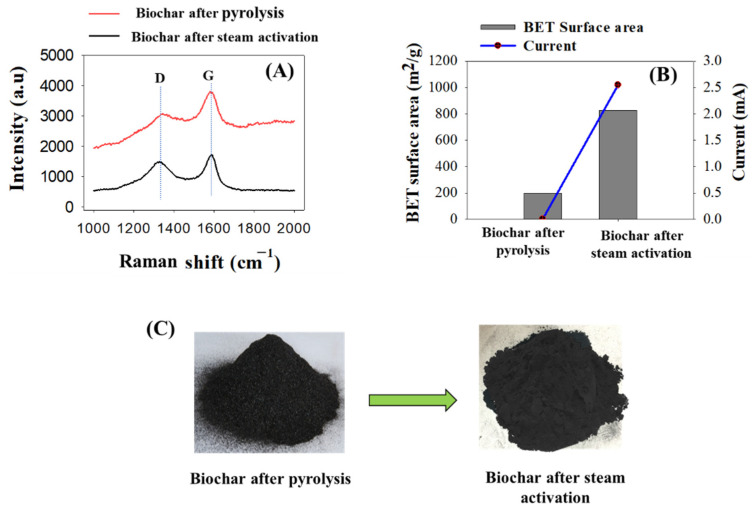
Physical and electrical properties analyzed for activated biochar compared to biochar sample. (**A**) Raman spectra (IR); (**B**) BET analysis and current measurement; (**C**) optical images of biochar after pyrolysis and biochar after steam activation.

**Figure 5 biosensors-12-00908-f005:**
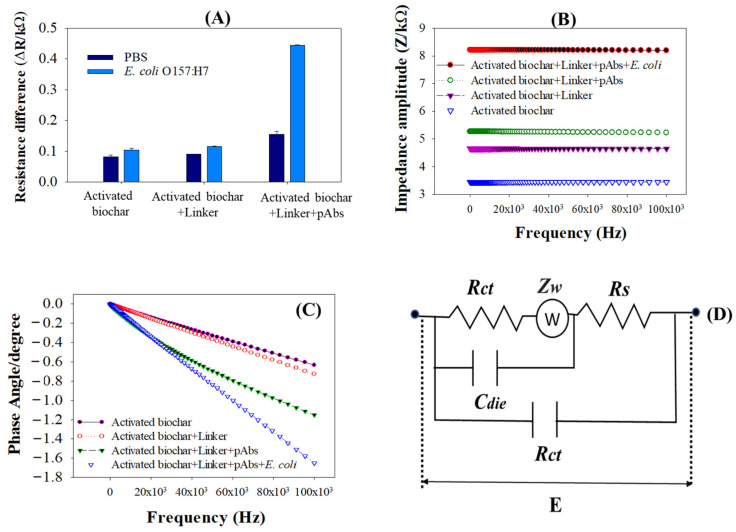
(**A**) Resistance difference (∆R) of activated biochar-based immunosensor reacted with PBS as control and *E. coli* O157:H7; (**B**) spectra of impedance amplitude; (**C**) spectra of phase angle after stepwise-immobilization of activated biochar, streptavidin as linker, anti-*E. coli* antibodies on the immunosensor, and their response to *E. coli* O157:H7; (**D**) an equivalent circuit based on electrochemical system of the immunosensor.

**Figure 6 biosensors-12-00908-f006:**
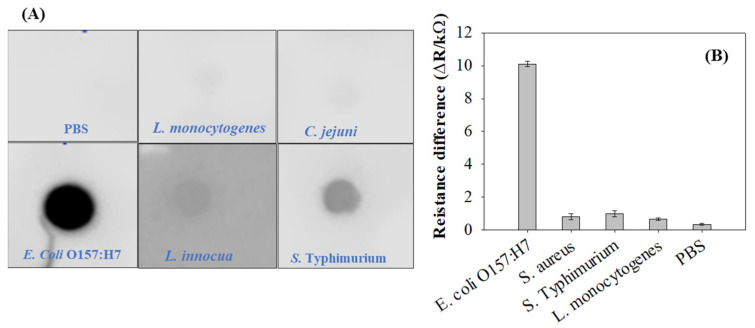
The specificity of custom-prepared anti-*E. coli* pAbs with selected bacterial strains determined by (**A**) dot blot assay; and (**B**) activated biochar-based immunosensor. PBS indicates phosphate-buffered saline as control.

**Figure 7 biosensors-12-00908-f007:**
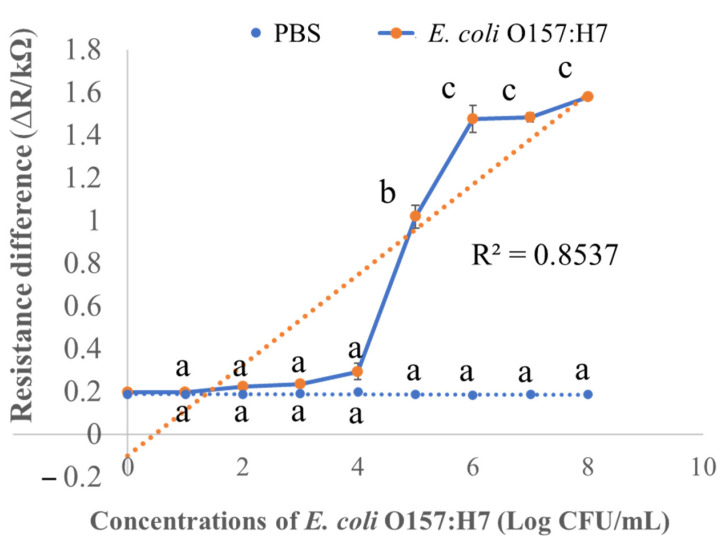
Resistance difference (∆R) of activated biochar-based immunosensor reacted with the selected concentration of *E. coli* O157:H7 from 10^1^ to 10^8^ CFU/mL. Different letters (a–c) denote a significant statistical difference among the concentrations at *p* < 0.05. PBS stands for phosphate-buffered saline as control.

**Figure 8 biosensors-12-00908-f008:**
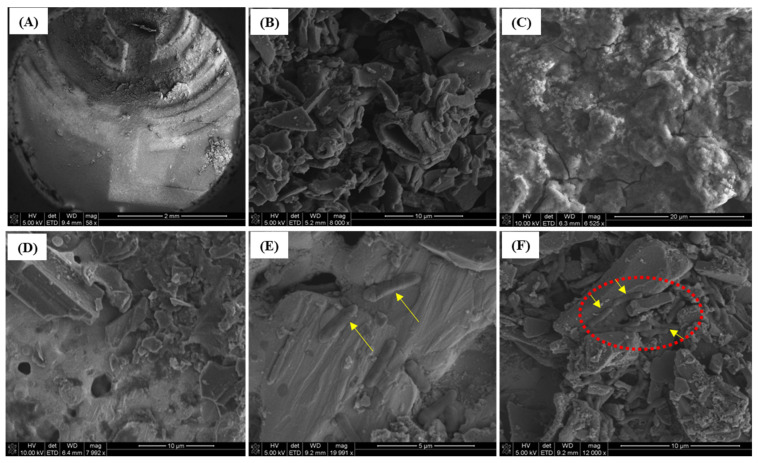
Scanning electron microscopic images of the developed immunosensor. (**A**) Activated biochar-based electrode with 2 mm scale bar; (**B**) activated biochar with 10 µm scale bar; (**C**) streptavidin-linker-modified electrode; (**D**) anti-*E. coli* antibody-immobilized biosensor reacted to PBS; (**E**) antibody-immobilized electrode with captured *E. coli* cells with 5 µm scale bar; (**F**) captured *E. coli* with 10 µm scale bar.

**Table 1 biosensors-12-00908-t001:** Optimization of different concentrations of activated biochar used for screen-printed electrode.

Concentration of Activated Biochar (mg/mL)	Resistance (kΩ)				
	1st Washing	2nd Washing	3rd Washing	4th Washing	5th Washing
5	NA	NA	NA	NA	NA
10	2.16 ± 0.0045 ^a^	1.93 ± 0.0047 ^b^	1.87 ± 0.0015 ^c^	1.81 ± 0.0011 ^c^	1.74 ± 0.002 ^c^
20	4.86 ± 0.007 ^d^	3.41 ± 1.28 ^d^	3.83 ± 0.005 ^d^	3.97 ± 0.031 ^d^	3.82 ± 0.239 ^d^
30	0.41 ± 0.0015 ^q^	0.403 ± 0.002 ^q^	0.32 ± 0.002 ^q^	0.386 ± 0.002 ^q^	0.381 ± 0.001 ^q^
40	0.98 ± 0.002 ^q^	0.684 ± 0.0036 ^q^	0.865 ± 0.004 ^q^	0.962 ± 0.003 ^q^	0.989 ± 0.0083 ^q^

Different superscript letters denote significant differences among the samples at *p* < 0.05. NA denotes not applicable as 5 mg/mL concentrations did not work for resistance change.

## Data Availability

Not applicable.

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
