# Peer review of "A Novel Activated Biochar-Based Immunosensor for Rapid Detection of E. coli O157:H7"

_biosensors, 2022, doi:10.3390/bios12100908_

Round 1

Reviewer 1 Report

This is an excellent research paper that reports the use of corn stover biochar linked with an antibody to detect E. coli O1557:H7.   E. coli O1557:H7 is one of the most important food pathogens.  Biochar is very economical to produce ($2000 a ton), and it is demonstrated that biochar can have similar properties to more expensive materials used for analytical detection of pathogens (graphene, carbon nanotubes).  

The introduction adequately covers the subject matter.  The experimental section explains in detail the assay and how the materials were prepared and used.  The results are presented at high scientific quality.  The results are discussed in adequate detail.  The conclusions are supported by the results.

Overall, it is a great paper that will be of much interest to analytical scientists and materials scientists.  Researchers working in Biochar will be especially interested in this manuscript.  However, there are some typos and misspellings in this manuscript that need to be addressed prior to publication.  It would be a shame if the article were published in its current form with the typos and misspellings.

Specific Comment to be addressed prior to publication: 

Specific comments to address:  In Figure 4.  The legend for the Raman plot states “Roman” which is incorrect.  This should be corrected.

The authors should take opportunity of a revision to carefully proofread their manuscript to catch similar errors.

Author Response

Responses to the comments from the editors and reviewers:

Reviewer #1: (Comments to the Author):

Comment 1: In Figure 4.  The legend for the Raman plot states “Roman” which is incorrect.  This should be corrected.

Answer: We have made the corrections in Fig. 4 and used “Raman” instead of “Roman” [Page-8, Fig. 4A].

Comment 2. The authors should take opportunity of a revision to carefully proofread their manuscript to catch similar errors.

Answer: Thanks for reviewer pointing out the typo. We carefully revised the manuscript and edited errors with red color throughout the manuscript.

Reviewer 2 Report

This article named “A novel activated biochar-based immunosensor for rapid detection of E. coli O157:H7” demonstrated that activated biochar-based immunosensor that can rapidly detect E. coli O157:H7 cells without incubation in pure culture. It may be considered for publication after careful modification.

1. In Figure 4C, there are only photo-images of biochar and activated biochar samples. There was no date for current.

2. A high solution SEM image of activated biochar maybe can provide a favorable evidence for enhanced surface area.

3. What is the ABC concentration in line-180. What is the purpose to suspend active biochar carbon into DMF solution? Is it okay to use a different suspension?

4. In line-277, the author described that temperature can play a key role in biochar activation. While the carbon is only activated at 800 ℃, how about other temperature, whether it has an impact on its structure?

5. Could you please explain more detail, the resistance values of the developed biosensor initially increased and then decreased as the concentration increased from 5 to 40 mg/mL. why the resistance is zero as the concentrate of activated biochar is aroud 5 mg/mL?

6. The minimum ΔR signal was noted for nontarget pathogens like S. aureus, S. Typhimurium, L. monocytogenes, L. innocua, do you test other bacilli as well as E. coli?

7. the scale is 2 mm (line-468) or 1 cm (line-438) in Figure 8a?

Author Response

Reviewer #: 2 (Comments to the Author):

Comment 1: 1. In Figure 4C, there are only photo-images of biochar and activated biochar samples. There was no date for current. 

Answer: The current values of sensor developments using raw biochar and steam activated biochar were depicted in Fig. 4B, whereas Fig.4C reveals photo-images of biochar powder before and after activation.

Comment 2: A high solution SEM image of activated biochar maybe can provide a favorable evidence for enhanced surface area.

Answer: Thanks for reviewer insight, we added a high-resolution biochar SEM image in Fig. 8B to provide more information regarding the activated biochar [Fig. 8B, Page-12, Line 445-448].

Comment 3: What is the ABC concentration in line-180. What is the purpose to suspend active biochar carbon into DMF solution? Is it okay to use a different suspension?

Answer: To optimize concentrations for an immunosensor development, we used several concentrations of ABCs ranging from 5 mg/ml to 40 mg/ml. The goal of suspending ABCs in DMF was to correctly sonicate and generate an ABCs slurry for senor fabrication. Although there are many different solvents available to suspend carbon particles for sonication, such as ethanol/water solvent, 1 M HCl solvent, etc., DMF is considerably easier to use and works better for proper sonication of ABCs.

Comment 4: In line-277, the author described that temperature can play a key role in biochar activation. While the carbon is only activated at 800 ℃, how about other temperature, whether it has an impact on its structure?

Answer: We deleted that sentence. In our prior study, biochar was activated between 300-1000 °C, and activated biochar was obtained for 800 ºC for 1 h when compared to others. In addition, the aim of this study was not to focus much on characterizations of biochar. Our next step of this work will be to further activate the biochar using a variety of activation time at higher temperature profiles in the propriety reactor to see if any change occurred in conductivity as you recommended.

Comment 5: Could you please explain more detail, the resistance values of the developed biosensor initially increased and then decreased as the concentration increased from 5 to 40 mg/mL. why the resistance is zero as the concentrate of activated biochar is around 5 mg/mL?

Answer: The resistance values of the developed biosensor initially increased and then decreased. At the concentration of 5 mg/ml, ABC concentrations did not hybridize the gaps between the electrodes on the sensor, which is why it did not show any electrical conductivity, which has been labeled as “not applicable or NA” in the Table 1 (we made modifications in Table 1). However, at 10 mg/ml of ABC concentrations, concentrations of ABC hybridized the electrode gaps, but bindings of ABCs to the sensor were minimal, which is why current flowing were interrupted over the sensor and increased resistance values. According to the Ohm's law, current is inversely proportional to the resistance.  At the concentration ranging from 20 to 40 mg/mL, the resistance values decreased due to higher concentrations and were less than 1 kW between 30-40 mg/ml because of unstable bindings of activated biochar to the sensor surface.

Comment 6: The minimum ΔR signal was noted for nontarget pathogens like S. aureus, S. Typhimurium, L. monocytogenes, L. innocua, do you test other bacilli as well as E. coli?

Answer: We primarily conducted the test employing a few non-targeted pathogens, such as L. monocytogenes, C. jejuni, S. typhimurium, L. innocua, S. aureus etc., to confirm antibody specificity when compared to E. coli O157:H7. As this was our first study on activated biochar-based immunosensors, the focus was to develop the biosensor with E. coli O157:H7. Our next step of this work is to perform the test extensively on biosensor using a variety of microorganisms and confirm the antibody specificity as you recommended.

Comment 7: the scale is 2 mm (line-468) or 1 cm (line-438) in Figure 8a?

Answer: The sentences associated with 1 cm scale bar (line-446) has been edited and changed with 2 mm [Page-12, Line 446].

Reviewer 3 Report

The Authors propose an activated biochar-based immunosensor that can detect E.coli without incubation in pure culture. The manuscript is well written and the results are very interesting. Therefore, I recommend the publication of the manuscript after addressing this minor comment.

- In order to improve the appeal of the manuscript, the Authors could also discuss on the AntiMicrobialResistance phenomenon that is very hot topic. In particular, other several devices have been proposed in literature to counteract this phenomenon (e.g,  Monitoring of individual bacteria using electro-photonic traps. Biomedical Optics Express10(7), 3463-3471, 2019; Rapid bacterial detection with an interdigitated array electrode by electrochemical impedance spectroscopy. Electrochimica Acta82, 126-131.2012; An integrated electro-optical biosensor system for rapid, low-cost detection of bacteria. Microelectronic Engineering239, 111523, 2021; A review on impedimetric immunosensors for pathogen and biomarker detection. Medical Microbiology and Immunology209(3), 343-362, 2020; Novel micro-nano optoelectronic biosensor for label-free real-time biofilm monitoring. Biosensors11(10), 361, 2021.). The performance of the proposed device could be also suitable for that goal.

Author Response

Reviewer #3: (Comments to the Author):  

Comment 1: In order to improve the appeal of the manuscript, the Authors could also discuss on the AntiMicrobialResistance phenomenon that is very hot topic. In particular, other several devices have been proposed in literature to counteract this phenomenon (e.g, Monitoring of individual bacteria using electro-photonic traps. Biomedical Optics Express, 10(7), 3463-3471, 2019; Rapid bacterial detection with an interdigitated array electrode by electrochemical impedance spectroscopy. Electrochimica Acta, 82, 126-131.2012; An integrated electro-optical biosensor system for rapid, low-cost detection of bacteria. Microelectronic Engineering, 239, 111523, 2021; A review on impedimetric immunosensors for pathogen and biomarker detection. Medical Microbiology and Immunology, 209(3), 343-362, 2020; Novel micro-nano optoelectronic biosensor for label-free real-time biofilm monitoring. Biosensors, 11(10), 361, 2021.). The performance of the proposed device could be also suitable for that goal. 

Answer: Thanks to the reviewer’s insight comment. We cited all suggested references of the interdigitated microelectrodes to the manuscript as you recommended [Page-2, Line 77-81].

Round 2

Reviewer 2 Report

the comments were answered clearly, I recommend to publish this work.